# SPiKE-SSM: A Sparse, Precise, and Efficient Spiking State Space Model for Long Sequences Learning

## Abstract

Spiking neural networks (SNNs) provide a low-power, energy-efficient solution by utilizing the spike-based and sparse nature of biological systems. Since the advent of Transformers, SNNs have struggled to compete with artificial networks on long sequential tasks, until the recent emergence of state space models (SSMs), which offer superior computational efficiency and modeling capability. However, applying the highly capable SSMs to SNNs for long sequences learning poses three major challenges: ❶ The membrane potential is determined by the past spiking history of the neuron, leading to reduced efficiency for sequence modeling in parallel computing scenarios. ❷ Complex dynamics of biological spiking neurons are crucial for functionality but challenging to simulate and exploit effectively in large networks. ❸ It is arduous to maintain high sparsity while achieving high accuracy for spiking neurons without resorting to dense computing, as utilized in artificial neuron-based SSMs. To address these challenges, we propose a sparse, precise and efficient spiking SSM framework, termed SPiKE-SSM. For ❶, we propose a boundary compression strategy (PMBC) to accelerate the inference of the spiking neuron model, enabling parallel processing for long sequence learning. For ❷, we propose a novel and concise neuron model incorporating reset-refractory mechanism to leverage the inherent temporal dimension for dynamic computing with biological interpretability. For ❸, we hierarchically integrate the proposed neuron model to the original SSM block, and enhance the dynamics of SPiKE-SSM by incorporating trainable thresholds and refractory magnitudes to balance accuracy and sparsity. Extensive experiments illustrate the effectiveness and robustness of SPiKE-SSM on the long range arena benchmarks and large language dataset WikiText-103, showing the potential of dynamic spiking neurons in efficient long sequence learning. The code will be publicly available.

## 1 Introduction

Spiking neural networks (SNNs) recently emerged as a competitive paradigm to improve AI energy efficiency. SNNs transmit information as binary spikes between synapses to perform sparse and event-driven computation. Despite being increasingly more competitive with artificial neural networks (ANNs) in vision tasks, SNNs still struggle with long-sequence modeling – a critical task for a wide range of temporal or sequential data-driven machine learning applications, such as text comprehending (Zhou et al., 2023), electroencephalograms spanning (Tang et al., 2023), etc.

Transformer (Vaswani et al., 2017) and its variants (Kitaev et al., 2020; Zaheer et al., 2020; Katharopoulos et al., 2020) have been developed for sequential tasks. However, their architectures are not suitable for SNN-based long sequence learning as SNN requires a time window-based simulation to enhance spike-based representation, resulting in slow inference compared to their ANN counterparts (Zhou et al., 2022; Yao et al., 2024). Moreover, the self-attention mechanisms (Vaswani et al., 2017) in Transformers are computationally intensive, contrasting with the energy-efficient properties of event-based representations and the sparse computation inherent to SNNs. As a competitive alternative to Transformer, state space models (SSMs) have garnered significant attention due to their long sequence modeling capabilities, such as S4 (Gu et al., 2021), DSS (Gupta et al., 2022), S5 (Smith et al., 2022) and Mamba (Gu & Dao, 2023). Notably, SSMs can achieve fast in-

Figure 1: Main ideas of SPikE-SSM for long-sequence modeling. (*Left*) **Overview:** A parallel max-min boundary compression (PMBC) strategy is proposed to address ❶ (§ 3.2); a new refractory neuron model with trainable dynamics is developed to address ❷ (§ 3.3). We integrate the proposed refractory neuron with *soft reset* within SSMs to address ❸ (§ 3.4). (*Right*) An example showing that the relevant information for the task at hand is often sparse in long-sequence inputs.

ference and parallel training by incorporating dynamic hidden states for handling long-range dependencies (LRDs), inspired by the low-complexity inference mechanism of recurrent neural networks (RNNs) (Sherstinsky, 2020; Schuster & Paliwal, 1997). Meanwhile, the sequential computing nature of SSMs is also more compatible with SNNs as the dynamics of spiking neurons can be inherently exploited in the temporal dimension. Furthermore, for tasks with long-sequence inputs, it is often the case that the *relevant information* to the problem at hand is *inherently sparse* (see Figure 1 *Right* for an example), aligning well with the sparse representation of SNNs.

Therefore, spiking SSMs naturally emerge as a promising paradigm for efficient long-sequence modeling. Recent works have highlighted notable advancements in capturing LRDs using spiking SSMs (Stan & Rhodes, 2023; Bal & Sengupta, 2024; Shen et al., 2024). However, these existing methods are still inadequate in addressing the following challenges when applying spike-based computation to SSMs: ❶ The membrane potential of a neuron in SNNs depends on its past spiking history, making parallel processing infeasible and, in turn, hindering the efficiency of sequence modeling. ❷ Biological neuron models exhibit complex dynamics that are essential for functionality (Urbanczik & Senn, 2014; Mikulasch et al., 2021; Capone et al., 2023) but challenging to simulate efficiently in large networks – an issue often overlooked by existing methods. ❸ Sparse representation is key for efficient computation in SNNs (Olshausen & Field, 2004; Jiao et al., 2022; Raposo et al., 2024); however, balancing the trade-off between sparsity (i.e., *spiking rate*) and accuracy remains challenging for spiking SSMs, as SSMs were originally designed on top of artificial neurons with dense computations.

In this work, we propose a novel spiking SSM model, termed **SP**ik**E**-SSM, to exploit the intricate dynamics of Leaky Integrate-and-Fire (LIF) neuron (Gerstner et al., 2014) in SSMs for **s**parse, **p**arallel, and **e**fficient long-sequence modeling. First, to address ❶, we propose a parallel max-min boundary compression strategy (PMBC) to accelerate the inference of the LIF neuron, enabling parallel processing for long sequence modeling. Second, to address ❷, we propose a refined LIF neuron model incorporating a reset-refractory dynamics to fully utilize the inherent temporal dimension for dynamic computing with biological interpretability; in the meantime, the hyperparameters of the proposed neuron model are trained efficiently and explicitly based on PMBC, enabling a systematic study of their functional impacts on the network. Third, to address ❸, we integrate the refractory neuron into an SSM block to adjust the membrane potential with dynamic reset, achieving both high accuracy and low spiking rate (i.e., high efficiency). An overview comparison of our SPikE-SSM with existing spiking SSMs is presented in Table 1. The main contributions are as follows:

- In this paper, we propose SPikE-SSM to effectively model the long sequence with SNNs. In contrast to existing spiking SSMs, our method can realize comprehensive parallel acceleration with trainable temporal dynamics, facilitating sparse, precise, efficient training and inference for long-range dependencies learning.

- To tackle the dilemma of event-driven neuronal dynamics with parallel processing for long sequence modeling, we propose a max-min boundary compression (PMBC) strategy to facilitate an efficient inference of SPikE-SSM. We empirically demonstrate that PMBC is versatile and effective for accelerating neuronal dynamics for parallel computing of SNNs.

- A new LIF neuron model with a refractory mechanism is proposed to fully utilize the inherent temporal dimension for biologically interpretable dynamic computation, achieving both high accuracy and sparsity with the trainable dynamics.

Table 1: Comparison of our model with existing spiking SSMs. Previous methods mostly apply binary activation to SSMs without considering the intricate neuronal dynamics. [†] SpikingSSM approximates the LIF neuron with hard reset dynamics by using a surrogate model, which is subject to approximation errors. [‡] SpikingSSM has partial trainable dynamics since hard reset is rough and simplified with limited dynamic variables. In contrast, our method can train the neuron hyperparameters and temporal dynamics efficiently and explicitly in a parallel manner with the proposed PMBC, enabling a functional study of their impact on the network. In SPikE-SSM, a more interpretable soft reset mechanism is employed, incorporating additional trainable dynamic variables and parameters.

| Method | Reset Mechanism | Trainable Dynamics |
|---|---|---|
| Binary-S4D (Stan & Rhodes, 2023) | ✗ | ✗ |
| S6-based SNN (Bal & Sengupta, 2024) | ✗ | ✗ |
| SpikingSSM (Shen et al., 2024) | ✓[†] | partial[‡] |
| SPikE-SSM (ours) | ✓ | ✓ |

- Extensive experiments are conducted on LRA benchmarks and the large-scale WikiText-103 language modeling databases, the results of which validate the effectiveness and efficiency of the proposed SPikE-SSM for long-range dependencies learning.

## 2 RELATED WORKS

### 2.1 LONG SEQUENCES LEARNING MODELS

Long sequence modeling has gained significant attention recently due to its widespread application across different domains such as text comprehending (Zhou et al., 2023), computer vision (Shi et al., 2024; Zhong et al., 2024) and electroencephalograms spanning (Tang et al., 2023). The key challenge in long-sequence modeling lies in efficiently compressing context into a manageable state while capturing information spread across observations separated by thousands of timesteps. To address them, Transformer and Attention (Vaswani et al., 2017; Dao et al., 2022; Dao, 2023) are proposed to retain the entire context during auto-regressive inference, which is effective but requires quadratic-time computational complexity. Although some Transformer variants (Kitaev et al., 2020; Katharopoulos et al., 2020) are proposed to reduce the compute and memory requirements, their performances on long-range reasoning remain considerably suboptimal (Gu et al., 2021). Inspired by RNNs, RWKV (Peng et al., 2023) combines the parallel training of transformers with the efficient inference of RNNs. Similarly, other recurrent models aim to compress context into a finite state, offering constant-time inference and linear-time training, but their effectiveness is limited by the quality of compression and a fixed representation space (Qin et al., 2023). More recently, SSM-based methods (Smith et al., 2022; Fu et al., 2022; Mehta et al., 2022) have emerged as a promising alternative to sequence models such as RNNs and Transformers. For example, HiPPO (Gu et al., 2020) pioneered compressing long inputs into dynamic representations using orthogonal polynomials, while S4 (Gu et al., 2021) advanced this with low-rank corrections for stable diagonalization and simplified Cauchy kernel operations. Mamba (Gu & Dao, 2023) focuses on selective state representations to optimize efficiency and effectiveness, using a selection mechanism and hardware-optimized algorithms to maintain robust contextual information capture. All above methods are based on artificial neurons with analog-valued output, resulting in dense vector-matrix multiplication (VMM) and huge computational costs. In contrast, the proposed SPikE-SSM utilizes the compatibility between the remarkable LRDs modeling ability of SSMs and the intrinsic dynamics of SNNs, promoting sparse training and fully parallel inference with trainable temporal dynamics.

### 2.2 SNNS-BASED SEQUENCE MODELING AND APPLICATIONS

SNNs (Ghosh-Dastidar & Adeli, 2009) have gained attention as a compelling bio-plausible and computational efficient substitute for traditional artificial neural networks (ANNs) in many vision tasks. However, SNNs have struggled to make significant progress in long-sequence modeling tasks due to the inherent serial computing nature. Therefore, to train SNNs in parallel, PSN (Fang et al., 2024) simplifies spiking neuron by omitting the reset mechanism, leading to reduced sparsity. To handle this issue, a probabilistic reset mechanism is proposed in PSU (Li et al., 2024) to achieve

parallel computing with elevated sparsity by decoupling the integration-spiking-resetting process, which comes at the expense of higher computational complexity. With the recent resurgence of SSMs, there has been a renewed focus on applying efficient parallel computing to SNNs. For example, SpikeS4 (Du et al., 2024) integrates LIF neurons with S4 layers for speech learning. Binary S4D builds a binary SSM by applying a spiking activation directly to the sum of hidden states, enabling parallel training but neglecting neuronal dynamics (Stan & Rhodes, 2023). To further enhance sparsity, a stochastic spiking neuron is proposed in S6-based SNN (Bal & Sengupta, 2024), which is trained with stochastic noises in gradients, resulting in accuracy degradation. More recently, SpikingSSMs (Shen et al., 2024) utilizes a surrogate dynamic network (SDN) to approximate the dynamics of LIF neurons, which extremely accelerates the training and inference by parallel computing. However, the pre-training requirement of SDN could constrain its application on more general dynamic spiking neurons which are hard to approximate. Due to the effectiveness of spike-based sequence learning, some SNNs-based language models are proposed for more efficient language modeling, such as SpikeGPT (Zhu et al., 2023) and SpikeBERT (Lv et al., 2023). In contrast to existing spiking SSMs, SPikE-SSM proposed in this paper realizes comprehensive parallel acceleration with trainable temporal dynamics, efficiently achieving both high sparsity and excellent accuracy for long-range dependencies learning, which possesses the potential and prospects for constructing low-energy language models and enabling widespread applications.

## 3 METHOD

### 3.1 PRELIMINARIES OF SSMS AND LIF NEURON

**SSMs.** According to (Gupta et al., 2022) and (Gu et al., 2021), SSMs provide a framework for long sequences modeling with lower computational complexity, which aims to transform an input sequence $x(t) = (x_0, \cdots, x_{L-1}) \in \mathbb{R}^{1 \times L}$ into an output sequence $y(t)(y_0, \cdots, y_{L-1}) \in \mathbb{R}^{1 \times L}$, where $L$ is the length of sequence. This transformation occurs with the aid of an implicit latent state $h(t) \in \mathbb{R}^{N \times 1}$, which captures the underlying dynamics and relationships between the input and output sequences. The continuous representation of this model is formulated as:

$$\frac{dh(t)}{dt} = h'(t) = Ah(t) + Bx(t), y(t) = Ch(t), \tag{1}$$

where the state matrix $A \in \mathbb{R}^{N \times N}$ and vectors $B \in \mathbb{R}^{N \times 1}$, $C \in \mathbb{R}^{1 \times N}$ are the parameters. To adapt SSM to real-world discrete data, one can discretize the continuous formulation Eq. (1) with discretization rules such as zero-order hold (Gupta et al., 2022; Voelker et al., 2019). Then $x(t)$ can be mapped to $y(t)$ in a recurrent view:

$$\bar{A} = e^{\Delta A}, \bar{B} = A^{-1}(\bar{A} - I)B, \bar{C} = C \implies h_t = \bar{A}h_{t-1} + \bar{B}x_t, y_t = \bar{C}h_t, \tag{2}$$

where $\Delta \in \mathbb{R}^+$ is the sample time, and $h_{-1} = 0$ for convenience. Note that the recurrence operation in Eq. (2) can be explicitly unrolled as a kernel view:

$$y_k = \sum_{j=0}^{k} \bar{K}_j \cdot x_{k-j}, \quad \bar{K} = \left(\overline{CB}, \overline{CAB}, \dots, \overline{CA}^{L-1}\bar{B}\right) \in \mathbb{R}^{1 \times L}, \tag{3}$$

which requires $\mathcal{O}(L^2)$ multiplications despite all the elements of $y$ can be expediently computed in parallel by computing the kernel $\bar{K}$ first. Fortunately, Eq. (3) can be accelerated by Fast Fourier Transform (FFT) (Duhamel & Vetterli, 1990) with time complexity $\mathcal{O}(L \log L)$ (Gupta et al., 2022).

**LIF Neuron.** The LIF neuron is widely used in spiking networks (Eshraghian et al., 2023), as it can capture the "leaky-integrate-fire-reset" process and balances ease of implementation with temporal dynamics by simplifying an RC circuit dynamical system (Gerstner et al., 2014). Let $t$ denote the time step, the input currents $I$ are linearly integrated into the membrane potential $u$ in LIF neuron, the process of which can be formulated as follows.

$$\tau \frac{du(t)}{dt} = -u(t) + IR, \quad u'_t = \beta u_{t-1} + (1-\beta)I_t, \quad s_t = H_s\left(u'_t - v_{\text{th}}\right), \tag{4}$$

where $\tau \in \mathbb{R}$ is the time constant and $\beta$ is its discrete-time equivalent. $R$ denotes the resistivity. $u'_t$ and $u_t$ are the membrane potentials before and after the trigger of a spike. $H_s$ denotes the the

---

**Algorithm 1** The Optimization Process of Parallel Max-min Boundary Compression (PMBC)

---

**Input:** Parameters $\tau, v_{\text{th}}, U_{\text{th}}$; Input signal $I \in \mathbb{R}^{1 \times L}$; Maximum of iterations $M$.
**Output:** Spiking signals $s \in \mathbb{R}^{1 \times L}$.
1: Define $p = (\tau^0, \tau^1, \cdots, \tau^{L-1})$; $k = \text{iFFT}(\text{FFT}(I) \cdot \text{FFT}(p))$.
2: Initialize $s^{up} = (1, \cdots, 1) \in \mathbb{R}^{1 \times L}$ and $s^{low} = (0, \cdots, 0) \in \mathbb{R}^{1 \times L}$.
3: **Repeat** up to $M$ times:
4:     $m^{up} = U_{\text{th}} \cdot \text{iFFT}(\text{FFT}(p) \cdot \text{FFT}(s^{up})) + v_{\text{th}}$;
5:     $m^{low} = U_{\text{th}} \cdot \text{iFFT}(\text{FFT}(p) \cdot \text{FFT}(s^{low})) + v_{\text{th}}$;
6:     **If** $k_t > m_t^{up}$, **then** $s_t^{low} = 1$; **If** $k_t < m_t^{low}$, **then** $s_t^{up} = 0$;
7: **Until** convergence of spike rate $\frac{1}{L} \sum_i s_i^{low}$.
8: **Return** $s = s^{low}$.

---

**Heaviside function of LIF.** As spikes are discrete events highly localized in time, a spike $s$ is emitted when the membrane potential exceeds the firing threshold ($v_{\text{th}} \in \mathbb{R}$), that is $s_t = 1$, otherwise $s_t = 0$. After firing, the membrane voltage is adjusted by the reset mechanism, making subsequent spiking more difficult. Specifically, the membrane voltage is either reset to a specific value $u_r$ (hard reset) or reduced by subtracting the same value $v_{\text{th}}$ as the firing threshold (soft reset), that is:

$$\text{soft reset: } u_t = u_t' - s_t v_{\text{th}}, \quad \text{hard reset: } u_t = u_t'(1 - s_t) + u_r. \tag{5}$$

From Eq. (5) we can observe that the hard reset clears all historical membrane voltage signals, while the soft reset retains a proportion of them after spiking, which is more bio-plausible. Furthermore, we creatively decouple the firing threshold value and soft reset magnitude into $v_{\text{th}}$ and $U_{\text{th}}$ respectively, which can promote the representation capability of LIF neuron. However, all the reset mechanisms introduce unavoidable iterative computations due to the form of temporal dependence and Heaviside function, similar to the nonlinearities in RNN.

## 3.2 PARALLEL MAX-MIN BOUNDARY COMPRESSION (PMBC)

This subsection aims to address Challenge ❶. According to discretizing the LIF neuron with the soft reset mechanism in Eq. (5) combined with a decoupling reset magnitude $U_{\text{th}}$, we can obtain the following formula:

$$u_t = \tau u_{t-1} - s_{t-1} U_{\text{th}} + I_t, \quad s_t = H_s(u_t - v_{\text{th}}). \tag{6}$$

The output membrane voltage $u$ is iteratively computed by Eq. (6) since $u_t$ depends on the spiking history from the previous time steps, notwithstanding the input current $I$ can be obtained in parallel. This leads to a significant reduction in computational efficiency, especially for long sequence inputs. To solve this problem, we propose the following assertion, which lays the foundation for subsequent parallel computation to accelerate training and inference (See Appendix A.1 for the proof).

**Assertion 3.1.** *The historical input signal $I$ and spiking information $s$ are deconstructed in the iteration process of Eq. (6), which is equivalent to:*

$$u_t = k_t - m_t + v_{\text{th}}, \quad s_t = H_s(k_t - m_t), \tag{7}$$

$$where \quad k_t = \sum_{i=1}^{t} \tau^{t-i} I_i, \quad m_t = U_{\text{th}} \sum_{i=1}^{t-1} \tau^{t-1-i} s_i + v_{\text{th}}. \tag{8}$$

Note that $\tau$ and $v_{\text{th}}$ are fixed in one training step. Given $I_i$ at all times, notice that the form of $k$ is the convolution of input sequence $I$ and the exponential sequence of $\tau$, we can obtain $k_t$ at all times in parallel with accelerated calculation through FFT. *The question of whether we can obtain $s_t$ in parallel becomes how to obtain $\text{m}_t$ at different times in parallel. To this end, we propose the PMBC strategy to address Challenge ❶.* It can be observed that the spiking signal $s$ is a binary variable, taking a value of either 0 or 1. Thus we can initialize the upper and lower bounds of $m_t^{up}$ and $m_t^{low}$ by setting all the spiking signals $s_i = 1, (i = 0, \cdots, L-1)$ and $s_j = 0, (j = 0, \cdots, L-1)$ respectively. The two bounds can be utilized to compare with $k_t$ simply, obtaining most spiking signals $s_t$ by parallel computation, and then update bounds values using these new $s_t$. This process can be iterated until convergence in order to obtain all spiking states as shown in Figure 2(a). We summarize

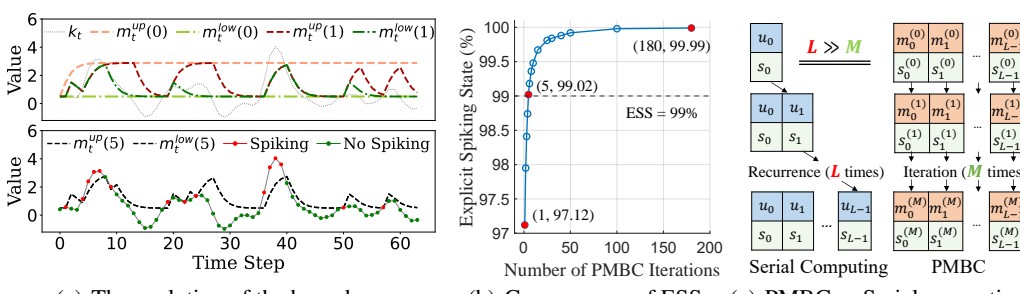

(a) The evolution of the boundary.   (b) Convergence of ESS.   (c) PMBC **vs** Serial computing.

Figure 2: Intuitive execution process of PMBC in Algorithm 2. ESS means explicit spiking state.

the process of parallel computation of PMBC as Algorithm 1. After finite iterations of PMBC, there may exist still a few fuzzy spiking signals $s_i$ unidentified, which can be assigned randomly or based on a prior distribution. The detailed discussion about the fuzzy spiking signals is provided in the Appendix C.3.2. To promote a lower spiking rate, we choose $s = s^{low}$ as the final output of spiking signals. To accelerate training and inference, we implement the FFT and inverse FFT operations with only setting $M = 3$, and the experimental results have proven that this configuration is capable of identifying around 99% of the spiking signals without compromising accuracy. Our method can determine the majority of spikes in the initial iterations, as shown in Figure 2(b). This is because the distribution of $k_t$ is closely tied to $I$, which is influenced by the normalization process before. With a proper initialization of $v_{th}$, the first PMBC iteration effectively identifies that most spiking signals are zero. This significantly reduces the number of required iterations and improve training efficiency (e.g., $M = 3$ vs. $L = 1024$). Figure 2(c) provides an intuitive comparison between traditional serial computing and PMBC. The detailed analysis of the boundary evolution and convergence process of PMBC in Figure 2 are described in the Appendix B.1. Particularly, we have the following assertion (see Appendix A.2 for proof):

**Assertion 3.2.** *For the input signal $y \in \mathbb{R}^{1 \times L}$, all the spiking signals can be identified with finite iterations of PMBC ($\leq L$), achieving significant acceleration compared to original serial computing.*

### 3.3 REFRACTORY LIF NEURON MODEL

In biological neurons, spiking is usually followed by a refractory period during which new spiking is more difficult. This mechanism improves the overall sparsity of the network and could substantially reduce its energy consumption. Therefore, to simulate the intrinsic temporal dynamics of realistic neurons and further improve network sparsity, we introduce an innovative refractory LIF neuron model based on the soft reset mechanism, which effectively *addresses Challenge* ❷. The LIF neuron with a refractory period can be mathematically described as:

$$u_t = \tau u_{t-1} + I_t - R_t U_{\text{th}}, \quad s_t = H_s(u_t - v_{\text{th}}), \tag{9}$$

$$where \quad R_t = \tau_r R_{t-1} + s_{t-1}, \tag{10}$$

In our refractory neuron model, $\tau_r$ is the refractory magnitude. $R_t$ denotes the refractory period-based sliding pulse, which is determined by both spiking signal $s_{t-1}$ and $R_{t-1}$ in the last time step. From Eq. (10) we can observe that the larger the value of the previous sliding pulse $R_{t-1}$, the greater $R_t$ becomes, causing membrane voltage $u_t$ to decrease accordingly, which makes it harder for the neuron to spike again during the refractory period. Similar to Assertion 3.1, we have the following results for the proposed refractory neuron model (see Appendix A.3 for the proof):

**Assertion 3.3.** *In the refractory LIF neuron, the historical input signal $I$ and spiking information $s$ is deconstructed in the iteration process of Eq. (9), which is equivalent to:*

$$u_t = k_t - m_t + v_{\text{th}}, \quad s_t = H_s(k_t - m_t), \tag{11}$$

$$where \quad k_t = \sum_{i=1}^{t} \tau^{t-i} I_i, \quad m_t = U_{\text{th}} \sum_{i=1}^{t-1} \sum_{j=0}^{t-1-i} (\tau/\tau_r)^j \tau_r^{t-1-i} s_i + v_{\text{th}}. \tag{12}$$

The PMBC algorithm of the refractory LIF neuron is summarized as Algorithm 2 in Appendix B.2, which only differs from the LIF neuron with soft reset in the representation of $m_t$.

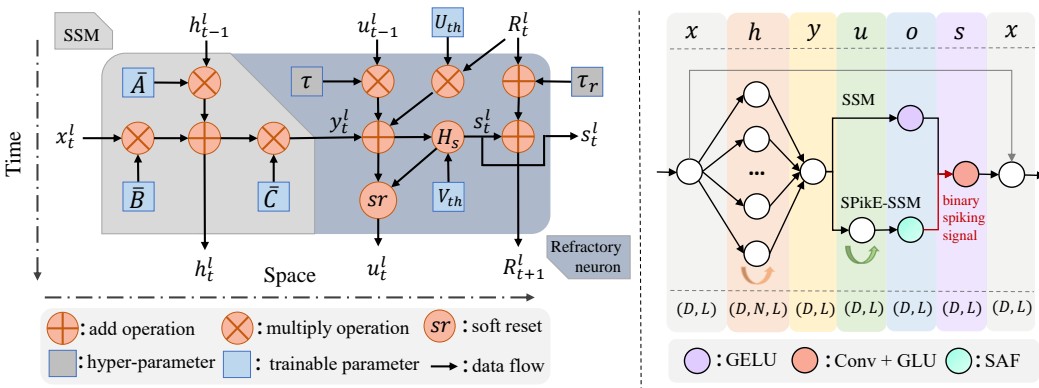

Figure 3: The SPikE-SSM block. **(Left)** Forward computation graph of a single SPikE-SSM layer. **(Right)** Comparison of SSMs. The original SSM outputs floating-point numbers, while SPikE-SSM replaces its non-linearity with the proposed refractory neuron model, which can incorporate higher-level neuronal dynamics for long sequence modeling. $D$, $N$, and $L$ represent the model dimension, SSM hidden dimension, and sequence length, respectively. SAF is the spiking activation function.

### 3.4 THE BLOCK OF SPikE-SSM

For Challenge ❸, due to the exceptional long sequence modeling capability of SSMs, we integrate the proposed refractory neuron with soft reset mechanism and PMBC to the inherent SSM block, which aims to maintain both the high sparsity and excellent accuracy in the inference progress. In the proposed SPikE-SSM, we choose the original block of S4D model (Gu et al., 2022) as the backbone since it can achieve pragmatic simplification to enhance model efficiency as the latest diagonal version of SSM. Then the output $y$ of the S4D block is activated by the proposed refractory neuron, hence Eq. (9) is rewritten as follows with Eq. (10) unchanged:

$$y_t = \bar{C}h_t, \quad u_t = \tau u_{t-1} + y_t - R_t U_{\text{th}}, \quad s_t = H_s(u_t - v_{\text{th}}). \tag{13}$$

Inspired from (Rathi & Roy, 2021), we render $v_{\text{th}}$ and $U_{\text{th}}$ as trainable parameters within the SPikE-SSM block. This approach is motivated by their pivotal role in regulating the neuron's spiking rate, thereby not only bolstering the SPikE-SSM's capability to attain exceptional performance and expedite the convergence of PMBC, but also serving as a further stepping stone to tackle Challenge ❸ with greater efficacy. The results of Eq. (13) are fed into a linear layer that comprises a Conv1D operation followed by a GLU activation function (Dauphin et al., 2017). The Conv1D enables efficient local feature extraction, while the GLU activation selectively gates the information flow, improving the model's ability to capture critical patterns in sparse binary data. Since the Heaviside function $H_s$ is non-differentiable at $x = 0$, we adopt the surrogate gradient (SG) method in SPikE-SSM. The details of SG in our method are provided in the Appendix B.3.

Figure 3 presents the forward computation graph of the SPikE-SSM block and the comparison with the original S4 blocks. Notably, from a neurobiological perspective, the SPikE-SSM block resembles a multi-time scale dendritic neuron (London & Häusser, 2005; Zheng et al., 2024), where $h$ represents the dendrites and $y$ the soma, both showing self-recurrent temporal dynamics.

## 4 EXPERIMENTS

In this section, we conduct extensive experiments to validate the superiority of our method, including the testing of long-range modeling capabilities on the sequential LRA and WikiText-103 tasks, with ablation studies and other related analyses. More experiments are shown in the Appendix C.

### 4.1 DATASETS AND EXPERIMENTAL SETTINGS

**Datasets.** In this paper, we perform experiments on extensive long sequence databases, including sequential MNIST (sMNIST) (Le et al., 2015), LRA benchmarks (comprising six tasks) (Tay et al., 2020) and WikiText-103 (one large Wikipedia text data) (Merity et al., 2016). The Details of these datasets are shown in the Appendix C.1.

Table 2: Accuracy performance comparison of SPikE-SSM and state-of-the-art methods on the LRA benchmarks. Since the original S4D-Lin failed on the Path-X task, we report the results of its close variant, S4D-Inv. Following S4D, we assume 50% accuracy for Path-X when not available and calculate the overall average (AVG) across all tasks. The best two results are highlighted in bold. For SpikingSSM and SPikE-SSM, the spiking rates (↓) of each task are highlighted in shaded gray areas. "—" indicates not applicable or unworkable, same for the other tables in this paper.

| Method | SNN | ListOps | Text | Retrieval | Image | Pathfinder | Path-X | AVG |
|---|---|---|---|---|---|---|---|---|
| Transformer (Vaswani et al., 2017) | No | 36.37 | 64.27 | 57.46 | 42.44 | 71.40 | — | 53.66 |
| LMUFormer (Liu et al., 2024) | No | 34.43 | 68.27 | 78.65 | 54.16 | 69.90 | — | 59.24 |
| S4D-Lin (Gu et al., 2021) | No | 60.52 | 86.97 | 90.96 | 87.93 | 93.96 | 92.80 | 85.52 |
| Spiking LMUFormer (Liu et al., 2024) | Yes | 37.30 | 65.80 | 79.76 | 55.65 | 72.68 | — | 60.20 |
| Binary S4D (Stan & Rhodes, 2024) | Yes | 54.80 | **82.50** | 85.03 | 82.00 | 82.60 | 61.20 | 74.69 |
| S6-based SNN (Bal & Sengupta, 2024) | Yes | 55.70 | 77.62 | 88.48 | 80.10 | 83.41 | — | 72.55 |
| SpikingSSM (Shen et al., 2024) | Yes | 59.93 | 82.35 | 88.20 | 86.81 | **93.68** | **94.80** | **84.30** |
| | | (13%) | (10%) | (6 %) | (22%) | (7 %) | (10%) | (11%) |
| SPikE-SSM (ours) | Yes | **60.17** | 82.43 | **88.82** | **87.23** | 92.04 | **94.37** | 84.18 |
| | | (12%) | (3 %) | (7 %) | (10%) | (9 %) | (7 %) | (8 %) |

Table 3: Perplexity performance comparison of SPikE-SSM with SOTA methods on WikiText-103. The symbol ↓ indicates that a smaller value for this metric is better, the same for other tables.

| Method | SNN | Perplexity (↓) | Parameters | Layer Count | Spiking Rate (↓) |
|---|---|---|---|---|---|
| Transformer (Vaswani et al., 2017) | No | 20.51 | 231M | 48 | — |
| S4 (Gu et al., 2021) | No | 20.95 | 249M | 48 | — |
| SpikeGPT (Zhu et al., 2023) | Yes | 39.75 | 213M | 48 | — |
| SpikingSSM (Shen et al., 2024) | Yes | 33.94 | 75M | 16 | 26.4% |
| SPikE-SSM (ours) | Yes | **33.18** | **75M** | **16** | **24.5%** |

**Implementation Details.** The hyper-parameters $\tau$ and $\tau_r$ are set to 0.1 and 0.9, respectively. To ensure the threshold and refractory magnitude are positive during training, the trainable parameters $v_{th}$ and $U_{th}$ are computed by $\exp(v_{th})$ and $\exp(U_{th})$ with zero initialization (i.e. $\exp(v_{th})$ and $\exp(U_{th})$ are initialized as 1). Other parameters of SPikE-SSM blocks are initialized same as S4D-Lin (Gu et al., 2022). SPikE-SSM is trained with Pytorch library on four NVIDIA A100-SXM4-80GB GPUs and AMD EPYC 7642 48-core CPUs, using AdamW optimization (Loshchilov, 2017). For sCIFAR10, sMNIST, psMNIST and LRA benchmarks, the model is trained by the cross-entropy loss (Mao et al., 2023) with accuracy (Acc) results reported, while the Perplexity results are reported for WikiText-103. The division of training and test data is consistent with (Shen et al., 2024). The details of settings on nine different tasks are described in Table 7 in Appendix C.2, including six LRA benchmarks, three sequential vision tasks, and a large text dataset (WikiText-103).

## 4.2 PERFORMANCES COMPARISONS

**Results on LRA Benchmarks.** Table 2 compares SPikE-SSM with both non-spiking and spiking networks using Transformer or SSM architectures . While maintaining accuracy comparable to the original model, SPikE-SSM achieves an average network sparsity of less than 10%. Additionally, our model shows a significant performance improvement over previous SNN sequence models. Notably, SPikE-SSM successfully tackles the Path-X task with extreme sparsity (only 0.07%). This task, which demands reasoning over long-range dependencies across sequences with 16,384 steps, is highly challenging and unsolvable by S4D-Lin, highlighting the robustness of our method.

**Results on WikiText-103.** In addition to LRA datasets, we further conduct experiments on the large Wikipedia text data, WikiText-103, to prove the advanced long sequence learning ability of SPikE-SSM against existing SOTA methods. The Perplexity results are shown in Table 3, which can be observed that SPikE-SSM achieves better performance with fewer parameters. Although model sparsity can improve computational efficiency, it is generally observed that achieving high accuracy often conflicts with maintaining strong sparsity, as sparsity typically results in information loss. However, it is particularly noteworthy that SPikE-SSM achieves both higher sparsity and accuracy compared to SpikingSSM, fully validating the effectiveness and superiority of the proposed model.

Table 4: Ablation studies of SPikE-SSM of different variants reported. Acc and SpkR denote Accuracy(%) ↑ and Spiking Rate(%) ↓ respectively. Spiking Rate is not applicable for ANN-S4D.

| Dataset | sMNIST | | psMNIST | | sCIFAR10 | |
|---|---|---|---|---|---|---|
| Criterion | Acc (%) | SpkR (%) | Acc (%) | SpkR (%) | Acc (%) | SpkR (%) |
| ANN-S4D | 99.50 | — | **98.20** | — | **87.11** | — |
| Spiking-S4D | 99.46 | 7.81 | 97.68 | 7.73 | 85.34 | 12.70 |
| SPikE-SSM-SR | 99.50 | 7.23 | 97.61 | 6.81 | 85.29 | 12.56 |
| SPikE-SSM-SRT | **99.51** | 6.09 | 96.97 | 5.65 | 85.61 | 11.03 |
| SPikE-SSM-SRR | 99.39 | **5.07** | 96.25 | **4.57** | 84.35 | **10.26** |
| **SPikE-SSM-Full** | **99.53** | 5.56 | 97.89 | 5.13 | 85.67 | 9.85 |

Table 5: Comparison of training speed of different methods. Training with the PMBC strategy achieves significant acceleration, the speed-up ratio amplifies with increasing sequence length.

| Method | Speed (iterations / s) ↑ | | | |
|---|---|---|---|---|
| | $L = 1K$ | $L = 2K$ | $L = 4K$ | $L = 8K$ |
| Training with BPTT (Mozer, 2013) | 0.60 | 0.29 | 0.11 | 0.03 |
| Training with SLTT (Meng et al., 2023) | 0.73 | 0.33 | 0.12 | 0.03 |
| Training with PMBC (ours) | **17.1** | **10.1** | **5.28** | **2.63** |
| Speed-up Ratio | 25.6× | 32.2× | 47.9× | 81.7× |

## 4.3 ABLATION STUDY

We conduct ablation studies to verify the design rationality of SPikE-SSM following the same experimental setups as Table 2. The variants with different levels of biological interpretability include:

- ANN-S4D. ANN-based SSM (S4D) model.
- Spiking-S4D. LIF-based spiking SSM without reset mechanism and refractory period.
- SPikE-SSM-SR. Only the soft reset mechanism is considered in the LIF neuron of SPikE-SSM block with PMBC, as shown in Eq. (6).
- SPikE-SSM-SRR. Both the soft reset mechanism and refractory period are considered in the LIF neuron of SPikE-SSM block with PMBC, as shown in Eq. (9-10).
- SPikE-SSM-SRT. Only the soft reset mechanism is considered in the LIF neuron of SPikE-SSM block with PMBC. $U_{th}$ and $V_{th}$ are trainable.
- SPikE-SSM-Full. Both the soft reset mechanism and refractory period are considered in the LIF neuron of SPikE-SSM block with PMBC. $U_{th}$ and $V_{th}$ are trainable.

Note that $U_{th}$ and $V_{th}$ are trainable only in SPikE-SSM-Full and SPikE-SSM-SRT. We compare the performances of different variants of SPikE-SSMs on sMNIST, psMNIST and sCIFAR10. The results are shown in Table 4, from which we can observe that each component designed for three Challenges is effective in SPikE-SSM. Specifically, the proposed refractory neuron model with the soft reset mechanism can optimize both high accuracy and pronounced sparsity with the thresholds $v_{th}$ and refractory magnitudes $U_{th}$ trainable in the SPikE-SSM block. More ablation studies about hyper-parameters $\tau$ and $\tau_r$, fire modes of fuzzy spiking signals, and the number of iterations $M$ in PMBC are shown in Tables 8 and 9 in the Appendix C.3.1, Tables 10 and 11 in the Appendix C.3.2, and Tables 12 and 13 respectively in the Appendix C.3.3, where our experiments in Figure 5 illustrates that SPikE-SSM with fixed $\tau$ and $\tau_r$ performs better than that with trainable $\tau$ and $\tau_r$.

## 4.4 TRAINING SPEED AND COMPUTATION COST ANALYSE

**The Superiority of PMBC on Training Speed.** We compare the training speed of SPikE-SSM, enhanced by our PMBC strategy, against traditional methods based on iterative LIF neurons, including Back-Propagation Through Time (BPTT) (Mozer, 2013) and the more recent Spatial Learning Through Time (SLTT) (Meng et al., 2023), which uses an optimized computational graph. The input consists of randomly generated 1-D sequences with various lengths of $L = 1K, 2K, 4K$, and $8K$,

Table 6: Computation cost comparison of SSM with ANN settings, SpikingSSM and SPikE-SSM on WikiText-103. "Ops" is an abbreviation for "operations".

| Model | Ops Types | Num of Ops (G) ↓ | Energy Cost (mJ) ↓ |
|-------|-----------|------------------|---------------------|
| SSM with ANN Settings | MAC | 275 | 1265 |
| SpikngSSM (Shen et al., 2024) | AC | 72.66 | 65.40 |
| SPikE-SSM | AC | **67.68** | **60.68** |

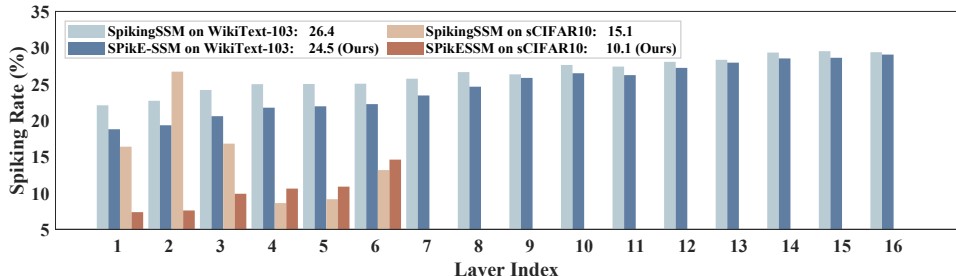

Figure 4: Spiking rate across all layers of SPikE-SSM and SpikingSSM on sCIFAR10 and WikiText-103 datasets. The number following each legend represents the respective average spiking rate.

and a batch size of 64. All time measurements were conducted on a single NVIDIA A100-SXM4-80GB GPU. As shown in Table 5, the speedup ratio using PMBC increases with sequence length, achieving a nearly two-order acceleration at $8K$.

**The Energy-efficiency of SPikE-SSM.** We compare the energy costs of the proposed SPikE-SSM and its corresponding ANN-based version on WikiText-103, the sequence length of which is $L = 8192$. Spiking networks are considered energy-efficient due to sparse binary activation. The multiplication between a binary activation and a floating-point weight can be performed using only addition operations in some neuromorphic chips (Yao et al., 2024). As a result, the primary operation in SNNs, synaptic accumulation (AC), incurs lower energy costs compared to the multiply-and-accumulate (MAC) operation in traditional ANNs. Although the hardware specifics and neuron dynamics are not considered here, a theoretical analysis can provide an estimate of SNN efficiency. Following previous studies (Yao et al., 2024; Li et al., 2024), we assume the energy cost of an MAC operation is $E_{\mathrm{MAC}} = 4.6pJ$, while an AC operation costs $E_{\mathrm{AC}} = 0.9pJ$ (Horowitz, 2014). In this part of the experiment, our model is set to comprise 16 layers, including a linear layer that projects spikes from $d = 1024$ to $d = 2048$. For specific quantitative comparison, we first report the spiking rates across different layers of SPikE-SSM in Figure 4. Then we report the MAC, AC, and energy consumption in these feature-mix layers since they occupy the majority of parameters and computations. Specifically, if these projections were fully computed via floating-point multiplications (SSM with ANN-based settings), they would require $275.2G$ MACs, consuming approximately $1.265J$. However, in our model (SPikE-SSM with SNN-based settings), the inputs to these layers are binary, with an average spiking rate of less than 25%. Based on the spiking rates in Figure 4, our model performs $67.42G$ ACs, consuming $60.68mJ$. The results are summarized in Table 6, which illustrate the high energy efficiency of SPikE-SSM compared with ANN-based SSM and SpikingSSM.

## 5 CONCLUSION

In this paper, we introduced SPikE-SSM, a novel spiking state space model designed to address key challenges in long-sequence learning with SNNs. Specifically, we innovatively address the conflict of event-driven neuronal dynamics with parallel computing in long sequence modeling by the PMBC method, enabling explicit and efficient training of neuronal dynamics. Subsequently, a concise reset-refractory neuron model is proposed to exploit the functionality of biological-plausible temporal dynamics. Its effective integration with the SSM block and incorporation of trainable thresholds and refractory magnitudes realize a balance between sparsity and accuracy. Extensive experiments on sequential vision tasks, LRA benchmarks, and WikiText-103 language modeling validate the superior efficiency, accuracy, and sparsity of SPikE-SSM. Our work shows the potential of dynamic spiking neurons in efficient long sequence learning.

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

# A  THE PROOFS

## A.1  THE PROOF OF ASSERTION 3.1

We utilize the Mathematical Induction Theory (Bather, 1994) to prove Assertion 3.1. As the presupposition, we set $s_0 = 0$ and $u_0 = 0$, which is reasonable since there's no signal at $t = 0$ with the time step of the spiking signal $s$, membrane voltage $u$, refractory momentum $R$ and input current $I$ signal all considered as 1 to $L$, that is $s_t$, $u_t$, $R_t$ and $I_t, t = 1, \cdots, L$. On this basis, according to Eq. (6) ($u_t = \tau u_{t-1} - s_{t-1} U_{\text{th}} + I_t, s_t = H_s(u_t - v_{\text{th}})$), we have the following derivation.

Firstly, consider $t = 1$, we can obtain:

$$u_1 = \tau u_0 - s_0 U_{\text{th}} + I_1 \xrightarrow{s_1 = H_s(u_1 - v_{\text{th}})} k_1 = I_1, \quad m_1 = v_{\text{th}}, \tag{14}$$

which is congruent to Eqs. (7-8) with $t = 1$. Then, to verify that Eqs. (7-8) hold for all $t \in [1, L]$, we assume that Eqs. (7-8) hold for $t = t_1 \in [1, L-1]$, that is

$$u_{t_1} = k_{t_1} - m_{t_1} + v_{\text{th}}, \quad s_{t_1} = H_s(k_{t_1} - m_{t_1}), \tag{15}$$

$$where \quad k_{t_1} = \sum_{i=1}^{t_1} \tau^{t_1 - i} I_i, \quad m_{t_1} = U_{\text{th}} \sum_{i=1}^{t_1 - 1} \tau^{t_1 - 1 - i} s_i + v_{\text{th}}, \tag{16}$$

based on this we have:

$$u_{t_1+1} = \tau u_{t_1} - s_{t_1} U_{\text{th}} + I_{t_1+1}, \quad s_{t_1+1} = H_s(u_{t_1+1} - v_{\text{th}}). \tag{17}$$

According to substituting the $u_{t_1}$ in Eq. (15) into Eq. (17), we have:

$$u_{t_1+1} = \tau(k_{t_1} - m_{t_1} + v_{\text{th}}) - s_{t_1} U_{\text{th}} + I_{t_1+1} \tag{18}$$

$$= \tau\left(\sum_{i=1}^{t_1} \tau^{t_1 - i} I_i - \left(U_{\text{th}} \sum_{i=1}^{t_1 - 1} \tau^{t_1 - 1 - i} s_i + v_{\text{th}}\right) + v_{\text{th}}\right) - s_{t_1} U_{\text{th}} + I_{t_1+1} \tag{19}$$

$$= \tau \sum_{i=1}^{t_1} \tau^{t_1 - i} I_i + I_{t_1+1} - U_{\text{th}} \sum_{i=1}^{t_1 - 1} \tau^{t_1 - 1 - i} s_i - s_{t_1} U_{\text{th}} \tag{20}$$

$$= \sum_{i=1}^{t_1+1} \tau^{t_1+1-i} I_i - \left(U_{\text{th}} \sum_{i=1}^{t_1} \tau^{t_1 - i} s_i + v_{\text{th}}\right) + v_{\text{th}} \tag{21}$$

$$= k_{t_1+1} - m_{t_1+1} + v_{\text{th}} \tag{22}$$

Therefore, Eqs. (7-8) hold for $t = t_1 + 1 \in [2, L]$. According to the Mathematical Induction Theory (Bather, 1994), we can conclude that Eqs. (7-8) hold for all the $t \in [1, L]$, so that Assertion 3.1 holds.                                                Q.E.D.

## A.2  THE PROOF OF ASSERTION 3.2

We assume that the dimension of input signal $I$ is $L$ for the refractory LIF neurons with the soft reset. Firstly, according to Eq. (7) and Eq. (8), we have

$$k_1 = I_1, \quad m_1 = v_{\text{th}}. \tag{23}$$

Hence we can definitely obtain $s_1 = H_s(k_1 - m_1)$ in the first iteration of PMBC. Then, without loss of generality, for the $(a+1)$-th iteration of PMBC, we assume that the first $a$ spiking signals (i.e. $s_1, s_2, \cdots, s_a$) have been obtained. Note that $m_{a+1}$ is only related to the first $a$ spiking signals:

$$k_{a+1} = \sum_{i=1}^{a+1} p_{a+1-i}^{a+1} y_i, \quad m_{a+1} = U_{\text{th}} \sum_{i=1}^{a} q_{a-i}^{a+1} s_i + v_{\text{th}}, \tag{24}$$

where $p^{a+1} = (\tau^0, \tau^1, \cdots, \tau^a)$, $q^{a+1} = (0, \tau^0, \tau^1, \cdots, \tau^{a-1})$. Therefore, in the $b$-th iteration of PMBC, at least one new spiking signal ($s_{a+1}$) can be definitely obtained by $s_{a+1} = H_s(k_{a+1} - m_{a+1})$, thus the maximum number of iterations of PMBC is no more than $L$.    Q.E.D.

In fact, due to the parallel computing mechanism of PMBC, the actual number of iterations is much smaller than $L$. As shown in Figure 2(b), after only 5 iterations, the explicit spiking state exceeds 99%, whereas processing a 1024-dimensional input sequence serially would require 1024 iterations, highlighting the efficiency of PMBC. Detailed experimental proofs and analysis are shown in Section B.1. Based on the proof process for Assertion 3.1, we can conclude that the distribution of $k_t$ is closely related to $y_t$, where $y_t$ is the output obtained from the previous SSM block, calculated through layer normalization. Therefore, if SPikE-SSM initializes $v_{th} = 1$ during training, the first iteration of PMBC can effectively determine that most of the spiking signals are zero. This significantly reduces the number of required iterations and accelerates training efficiency.

### A.3 THE PROOF OF ASSERTION 3.3

Similarly to the proof of Assertion 3.1, we utilize the Mathematical Induction Theory (Bather, 1994) to proof Assertion 3.3. As the presupposition, we set $s_0 = 0$, $u_0 = 0$, and $R_0 = 0$, which is reasonable since there's no signal at $t = 0$ with the time step of the spiking signal $s$, membrane voltage $u$ and input current $I$ signal all considered as 1 to $L$, that is $s_t$, $u_t$, and $I_t$, $t = 1, \cdots, L$. On this basis, according to Eqs. (11-12) ($u_t = \tau u_{t-1} + I_t - R_t U_{th}$, $R_t = \tau_r R_{t-1} + s_{t-1}$, $s_t = H(u_t - v_{th})$), we can transform the problem into proving the following equation first.

$$u_t = k_t - m_t + v_{th}, \quad s_t = H_s(k_t - m_t), \tag{25}$$

$$where \quad k_t = \sum_{i=1}^{t} \tau^{t-i} I_i, \quad m_t = U_{th} \sum_{i=1}^{t-1} \tau^{t-1-i} R_{i+1} + v_{th}, \quad R_{t+1} = \tau_r R_t + s_t, \tag{26}$$

Firstly, consider $t = 1$, we can obtain:

$$u_1 = \tau u_0 - R_1 U_{th} + I_1 \xrightarrow[R_1 = \tau_r R_0 + s_0]{s_1 = H_s(u_1 - v_{th})} k_1 = I_1, \quad m_1 = v_{th}, \tag{27}$$

which is congruent to Eq. (26) with $t = 1$. Then, to verify that Eqs. (25-26) hold for all $t \in [1, L]$, we assume that Eqs. (25-26) hold for $t = t_1 \in [1, L-1]$, that is

$$u_{t_1} = k_{t_1} - m_{t_1} + v_{th}, \quad s_{t_1} = H_s(k_{t_1} - m_{t_1}), \tag{28}$$

$$where \quad k_{t_1} = \sum_{i=1}^{t_1} \tau^{t_1-i} I_i, \quad m_{t_1} = U_{th} \sum_{i=1}^{t_1-1} \tau^{t_1-1-i} R_{i+1} + v_{th}, \quad R_{t_1+1} = \tau_r R_{t_1} + s_t, \tag{29}$$

based on this we have:

$$u_{t_1+1} = \tau u_{t_1} - R_{t_1+1} U_{th} + I_{t_1+1}, \quad s_{t_1+1} = H_s(u_{t_1+1} - v_{th}). \tag{30}$$

According to substituting the $u_{t_1}$ in Eq. (28) into Eq. (30), we have:

$$u_{t_1+1} = \tau(k_{t_1} - m_{t_1} + v_{th}) - R_{t_1+1} U_{th} + I_{t_1+1} \tag{31}$$

$$= \tau \left( \sum_{i=1}^{t_1} \tau^{t_1-i} I_i - \left( U_{th} \sum_{i=1}^{t_1-1} \tau^{t_1-1-i} R_{i+1} + v_{th} \right) + v_{th} \right) - R_{t_1+1} U_{th} + I_{t_1+1} \tag{32}$$

$$= \tau \sum_{i=1}^{t_1} \tau^{t_1-i} I_i + I_{t_1+1} - U_{th} \sum_{i=1}^{t_1-1} \tau^{t_1-1-i} R_{i+1} - R_{t_1+1} U_{th} \tag{33}$$

$$= \sum_{i=1}^{t_1+1} \tau^{t_1+1-i} I_i - \left( U_{th} \sum_{i=1}^{t_1} \tau^{t_1-i} R_{i+1} + v_{th} \right) + v_{th} \tag{34}$$

$$= k_{t_1+1} - m_{t_1+1} + v_{th} \tag{35}$$

Therefore, Eqs. (25-26) hold for $t = t_1 + 1 \in [2, L]$. According to the Mathematical Induction Theory (Bather, 1994), we can conclude that Eqs. (25-26) hold for all the $t \in [1, L]$. Subsequently, the parallel calculation form of the membrane potential of the LIF neuron with soft reset and refractory

period can be expressed as:

$$u_t = \sum_{i=1}^{t} \tau^{t-i} I_i - U_{\text{th}} \sum_{i=1}^{t} \tau^{t-i} R_i \tag{36}$$

$$= \sum_{i=1}^{t} \tau^{t-i} I_i - U_{\text{th}} \sum_{i=1}^{t} \tau^{t-i} (\sum_{j=1}^{i-1} \tau_r^{i-1-j} s_j) \tag{37}$$

$$= \sum_{i=1}^{t} \tau^{t-i} I_i - U_{\text{th}} \sum_{i=1}^{t-1} \sum_{j=0}^{t-1-i} \tau^j \tau_r^{t-1-i-j} s_i \tag{38}$$

$$= \sum_{i=1}^{t} \tau^{t-i} I_i - U_{\text{th}} \sum_{i=1}^{t-1} \sum_{j=0}^{t-1-i} (\tau/\tau_r)^j \tau_r^{t-1-i} s_i \tag{39}$$

$$s_t = H_s(u_t - v_{\text{th}}), \quad k_t = \sum_{i=1}^{t} \tau^{t-i} I_i \tag{40}$$

$$m_t = U_{\text{th}} \sum_{i=1}^{t-1} \sum_{j=0}^{t-1-i} (\tau/\tau_r)^j \tau_r^{t-1-i} s_i + v_{\text{th}} \tag{41}$$

Therefore, Assertion 3.3 holds. Q.E.D.

Apparently, Eqs. (11-12) degrade to the version of SPikE-SSM-RS without the refractory period when $\tau_r = 0$.

# B MORE DETAILS RELATED TO SPikE-SSM

## B.1 DETAILED DESCRIPTION AND ANALYSIS FOR FIGURE 2

To clearly demonstrate the implementation process and effectiveness of PMBC, we present the evolution of the boundary and convergence process in Figure 2.

(1) In Figure 2(a), we illustrate the boundary's evolution of a particular neuron chosen at random during training as PMBC iterations increase over different time steps. In the **top part of Figure 2(a)**, before the first iteration, with all spiking signals $s_i$ setting to 1, $m_t^{up}(0)$ increases slowly as the time step $t$ progresses, with growth gradually leveling off until stabilizing. This is consistent with Eq. (8), as $0 < \tau < 1$ causes $\tau^t$ to approach 0 over time, $m_t^{up}(0)$ approaches $U_{\text{th}}/(1-\tau) + v_{\text{th}}$ when $t$ is large. Conversely, with all spiking signals $s_i$ setting to 0, $m_t^{low}(0)$ remains $v_{\text{th}}$ across all time steps. After the first iteration of PMBC, partial binary spiking signals $s_i$ are explicitly obtained, leading to two outcomes: in some time steps, $m_t^{low}(1)$ apparently increases compared to $m_t^{low}(0)$, while in others, $m_t^{up}(1)$ apparently decreases compared to $m_t^{up}(0)$. It is observed that spiking states that are quickly identified are often following several $k_t$ that are too large or too small in succession. In the **bottom part of Figure 2(a)**, inherited from the convergence process of $m_t^{up}$ and $m_t^{low}$ in the top part, the results after 5 iterations of PMBC are shown, where $m_t^{low}(5)$ and $m_t^{up}(5)$ become almost identical. Eventually, nearly all binary spiking signals $s_i$ are explicitly determined based on $s_t = H_s(k_t - m_t)$, by comparing $k_t$ and $m_t$ in parallel and efficiently, omitting the serial computing of membrane potential $u_t$.

(2) In Figure 2(b), we show the convergence curve of the explicit spiking state as PMBC iterations increase. The explicit spiking state refers to the proportion of spiking signals that are explicitly determined across all time steps. From Figure 2(b), it is evident that PMBC can resolve most of the spikes in just a few iterations, significantly fewer than the original serial computation method used for LIF neurons (Eq. (4)). After only 5 iterations, the explicit spiking state exceeds 99%, whereas processing a 1024-dimensional input sequence serially would require 1024 iterations, highlighting the efficiency of PMBC.

(3) In Figure 2(c), we provide an intuitive comparison between the PMBC method based on parallel computing and the traditional serial computation method. For a sequence of length $L$, the serial method requires $L$ iterations, whereas the PMBC approach processes all $L$ tokens simultaneously in parallel, requiring only $M$ iterations, with $M$ much smaller than $L$.

---

**Algorithm 2** The Optimization Process of PMBC for the Refractory LIF Neuron

---

**Input:** Parameters $\tau, \tau_r, U_{\text{th}}$ and $v_{\text{th}}$; Input signal $I \in \mathbb{R}^{1 \times L}$; Maximum of iterations $M$.
**Output:** Spiking signals $s \in \mathbb{R}^{1 \times L}$.
1: Define $p = (\tau^0, \tau^1, \cdots, \tau^{L-1})$, $q : q_t = \tau_r^t \cdot \sum_{j=0}^{t}(\tau/\tau_r)^j$; $k = \text{iFFT}(\text{FFT}(I) \cdot \text{FFT}(p))$.
2: Initialize $s^{up} = (1, \cdots, 1) \in \mathbb{R}^{1 \times L}$ and $s^{low} = (0, \cdots, 0) \in \mathbb{R}^{1 \times L}$.
3: **Repeat** up to $M$ times:
4:    $m^{up} = U_{\text{th}} \cdot \text{iFFT}(\text{FFT}(q) \cdot \text{FFT}(s^{up})) + v_{\text{th}}$;
5:    $m^{low} = U_{\text{th}} \cdot \text{iFFT}(\text{FFT}(q) \cdot \text{FFT}(s^{low})) + v_{\text{th}}$;
6:    **If** $k_t > m_t^{up}$, **then** $s_t^{low} = 1$; **If** $k_t < m_t^{low}$, **then** $s_t^{up} = 0$;
7: **Until** convergence of spike rate $\frac{1}{L}\sum_i s_i^{low}$.
8: **Return** $s = s^{low}$.

---

In fact, the parallel PMBC method makes explicitly training parametric LIF neurons more efficient and feasible, especially in long sequence scenarios. In contrast, the SDN-based approach in SpikingSSM (Shen et al., 2024) is limited to effectively training only the threshold voltage ($v_{th}$), while it's unclear whether other hyper-parameters can be trained end-to-end. This ambiguity highlights the advantage of PMBC in handling more comprehensive parameter optimization with trainable temporal dynamics.

The experimental setup for this part is described as follows:

For Figure 2(a): Both the soft reset mechanism and refractory period are considered in the LIF neuron of SPikE-SSM block trained with PMBC. The parameters $\tau$, $\tau_r$, $U_{th}$, and $v_{th}$ are fixed as 0.3, 0.4, 1, and 0.5 respectively. The number of iterations in PMBC is set to 10 during training. The curve in Figure 2(a) shows the results with different iteration numbers in PMBC during inference.

For Figure 2(b): Only the soft reset mechanism is considered in the LIF neuron of the SPikE-SSM block trained with PMBC. The parameters $\tau$, $U_{th}$, and $v_{th}$ are fixed as 0.1, 1, and 3 respectively. The number of iterations in PMBC is set to 10 during training. After all the iterations of PMBC, the still fuzzy spiking signals are set to 1 by default, i.e. the spiking signals for these time steps are set to fire by default. The curve in Figure 2(b) shows the explicit spiking state with different iteration numbers in PMBC during inference.

## B.2 THE OPTIMIZATION PROCESS OF PMBC FOR THE REFRACTORY LIF NEURON

The pseudo-code of the optimization process of PMBC for the Refractory LIF Neuron is summarized in Algorithm 2, which only differs from the LIF neuron with soft reset in the representation of $m_t$ in Algorithm 1.

## B.3 SURROGATE GRADIENT IN SPikE-SSM

Since the Heaviside function $H_s$ in Eq. (13) is non-differentiable at $x = 0$, several surrogate gradient (SG) methods are proposed to enable training through gradient descent. Common SG functions are differentiable at all points and possess non-zero derivatives near the threshold, allowing them to approximate the original discontinuous gradient of the spiking activation function, such as the rectangular function (Zheng et al., 2021) and the triangular function (Bellec et al., 2018). In SPikE-SSM, the piecewise quadratic surrogate spiking function $g(x)$ is utilized with $\alpha = 1$. $g(x)$ and its gradient $g'(x)$ are defined as:

$$g(x) = \begin{cases} 0, & \text{if } x < -\frac{1}{\alpha} \\ -\frac{1}{2}\alpha^2|x|x + \alpha x + \frac{1}{2}, & \text{if } |x| \leq \frac{1}{\alpha} \\ 1, & \text{if } x > \frac{1}{\alpha} \end{cases}, \quad g'(x) = \begin{cases} 0, & \text{if } |x| > \frac{1}{\alpha} \\ -\alpha^2|x| + \alpha, & \text{if } |x| \leq \frac{1}{\alpha} \end{cases}. \quad (42)$$

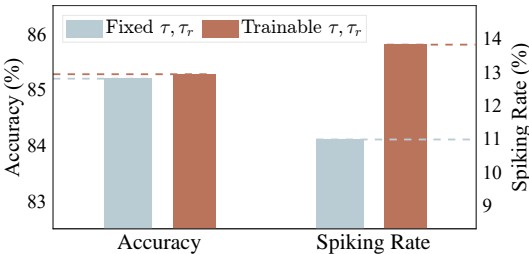

Figure 5: Comparison on sCIFAR10 between SPikE-SSM-SRR with fixed $\tau$ and $\tau_r$ and that with trainable $\tau$ and $\tau_r$. $U_{th}$ and $U_{th}$ are both set to 1 in SPikE-SSM-SRR.

Table 7: The hyper-parameters of our experiments on these datasets. **H** denotes the model dimension, **N** denotes the state dimension, **LR** denotes learning rate, **WD** denotes weight decay and **BS** denotes the batch size. **BN** and **LN** refer to Batch Normalization and Layer Normalization.

| Dataset | Depth | H | N | Norm | pNorm | Dropout | LR | BS | Epochs | WD | $(\Delta_{min}, \Delta_{max})$ |
|---------|-------|------|----|------|-------|---------|--------|----|--------|------|------------------------|
| sMNIST | 2 | 128 | 64 | LN | False | 0.1 | 0.01 | 64 | 25 | 0.01 | (0.001,0.1) |
| psMNIST | 4 | 128 | 64 | LN | False | 0.1 | 0.01 | 64 | 60 | 0.01 | (0.001,0.1) |
| sCIFAR10 | 4 | 128 | 64 | LN | False | 0.1 | 0.01 | 64 | 100 | 0.01 | (0.001,0.1) |
| ListOps | 8 | 128 | 64 | BN | False | 0 | 0.01 | 50 | 40 | 0.05 | (0.001, 0.1) |
| Text | 6 | 256 | 64 | BN | True | 0 | 0.01 | 16 | 32 | 0.01 | (0.001, 0.1) |
| Retrieval | 6 | 256 | 64 | BN | True | 0 | 0.01 | 64 | 20 | 0.01 | (0.001, 0.1) |
| Image | 6 | 512 | 64 | LN | False | 0.1 | 0.01 | 50 | 200 | 0.01 | (0.001, 0.1) |
| Pathfinder | 6 | 256 | 64 | BN | True | 0 | 0.004 | 64 | 200 | 0.01 | (0.001, 0.1) |
| Path-X | 6 | 256 | 64 | BN | True | 0 | 0.0005 | 32 | 50 | 0.01 | (0.0001, 0.01) |
| WT-103 | 16 | 1024 | 64 | LN | True | 0.1 | 0.0005 | 1 | 200 | 0.01 | (0.001,0.1) |

## C  MORE DETAILS RELATED TO EXPERIMENTS

### C.1  DETAILS OF DATASETS

#### C.1.1  SEQUENTIAL VISION DATASETS

The MNIST dataset (Deng, 2012) is a classic benchmark in machine learning, featuring 70,000 grayscale images of handwritten digits (0-9), with 60,000 samples for training and 10,000 for testing, each image sized at 28×28 pixels. It has long been a cornerstone for evaluating image classification models due to its simplicity and widespread availability. The sequential MNIST (sMNIST) dataset (Le et al., 2015) transforms these 2D images into sequences of 784 elements by flattening the pixel grid into a 1D sequence. This transformation poses a more complex challenge, as models must process and retain information over longer time steps to accurately classify the digit. To further increase the difficulty, the permuted sequential MNIST (psMNIST) (Le et al., 2015) applies a fixed random permutation to the pixel sequences, effectively scrambling their spatial order and disrupting any inherent structure. This variant demands even greater computational ability from models, as they must learn to extract meaningful features from a sequence with no obvious temporal or spatial coherence, making psMNIST a far more challenging task compared to the original sMNIST. The sCIFAR10 dataset is a sequential version of the CIFAR-10 dataset (Krizhevsky et al., 2009), where the original static images are processed in a temporal manner, typically by feeding the image pixels row-by-row or in a pre-defined sequence.

#### C.1.2  LRA BENCHMARK

The LRA benchmark (Tay et al., 2020) was specifically designed to evaluate the performance of sequence models in long-context scenarios, where capturing dependencies across extended sequences is crucial. It consists of six diverse tasks, with sequence lengths ranging from $1K$ to $16K$ steps, covering multiple modalities including visual data, mathematical expressions, and natural language. These tasks are carefully curated to challenge models on various aspects of long-context comprehensions, such as text classification, document retrieval, image recognition, the pathfinder problem,

Table 8: Ablation studies of SPiKE-SSM-SR ($v_{th}$ and $U_{th}$ are trainable) with different $\tau$. Acc, SpkR and FzR denote test accuracy, spiking rate and fuzzy rate respectively.

| Iter Nums | $\tau$ | 0.2 | 0.15 | 0.1 |
|---|---|---|---|---|
| 5 | Acc (%) ↑ | 84.91 | 84.93 | 84.93 |
| | SpkR (%) ↓ | 10.66 | 10.69 | 9.91 |
| | FzR (%) ↓ | 1.88 | 1.29 | 1.30 |
| 20 | Acc (%) ↑ | 85.24 | 84.37 | 85.12 |
| | SpkR (%) ↓ | 10.03 | 9.72 | 9.98 |
| | FzR (%) ↓ | 0.48 | 0.36 | 0.37 |

Table 9: Ablation studies of SPiKE-SSM-Full with different $\tau$ and $\tau_r$. Acc, SpkR and FzR denote test accuracy, spiking rate and fuzzy rate respectively. Fuzzy rate is defined as the mean proportion of final unidentified spiking signals $s_i$ in all neurons after finite iterations of PMBC during inference.

| $\tau$ | | 0.2 | | | 0.15 | | | 0.1 | | |
|---|---|---|---|---|---|---|---|---|---|---|
| Iter Nums | $\tau_r$ | 0.6 | 0.75 | 0.9 | 0.6 | 0.75 | 0.9 | 0.6 | 0.75 | 0.9 |
| 5 | Acc (%) ↑ | 84.97 | 84.60 | 83.32 | 84.13 | 84.34 | 84.75 | 85.17 | 85.04 | 84.37 |
| | SpkR (%) ↓ | 10.27 | 10.23 | 10.21 | 9.93 | 9.94 | 9.94 | 9.46 | 9.63 | 9.61 |
| | FzR (%) ↓ | 2.54 | 3.09 | 6.25 | 2.03 | 2.27 | 5.74 | 2.00 | 2.39 | 5.82 |
| 20 | Acc (%) ↑ | 84.64 | 84.56 | 83.32 | 84.25 | 84.17 | 83.85 | 84.66 | 84.47 | 83.56 |
| | SpkR (%) ↓ | 10.28 | 10.55 | 10.73 | 10.41 | 10.39 | 10.22 | 10.10 | 10.00 | 10.33 |
| | FzR (%) ↓ | 1.07 | 1.64 | 3.05 | 0.94 | 1.21 | 3.1 | 0.95 | 1.18 | 2.87 |

and list operations (ListOps), making it a comprehensive testbed for assessing a model's ability to process and reason over extended input sequences.

### C.1.3 WIKITEXT-103

The WikiText-103 dataset (Merity et al., 2016) is a large-scale corpus containing over 100 million tokens extracted from Wikipedia articles that have been rated as Good or Featured. Spanning a broad spectrum of topics and domains, it offers a rich variety of linguistic patterns and structures. Unlike many other datasets, WikiText-103 consists of full-length articles rather than isolated snippets, making it particularly well-suited for models designed to capture long-term dependencies across extended contexts. Due to its depth and diversity, it has become a pivotal benchmark for word-level language modeling, providing a robust testing ground for evaluating models' capacity to understand and generate coherent text over lengthy sequences.

### C.2 DETAILS OF EXPERIMENTAL SETTINGS

Table 7 describes the particular training details of experiments on different tasks, where LRA benchmarks consist of six tasks, including ListOps, Text, Retrieval, Image, Pathfinder and Path-X. **Depth** denotes the number of SPiKE-SSM blocks, $pNorm$ denotes the pre-norm, and $WT-103$ means WikiText-103 dataset.

### C.3 MORE EXPERIMENTAL RESULTS

### C.3.1 PARAMETER SENSITIVITY ANALYSIS

In the blocks of the proposed model, we compare the performances of SPiKE-SSM-SSR with fixed $\tau$ and $\tau_r$ and that with trainable $\tau$ and $\tau_r$. The results are shown in Figure 5, which shows that SPiKE-SSMwith fixed $\tau$ and $\tau_r$ can achieve lower sparsity and higher accuracy. Therefore, we $\tau$ and $\tau_r$ are set fixed in our method. Then we investigate the impacts of different $\tau$ and $\tau_r$ on our model.

Table 10: The impact of different fire modes for the fuzzy spiking signals on SPikE-SSM-SR with fixed $U_{\mathrm{th}} = 1$ and $V_{\mathrm{th}} = 1$.

| Criterion | No Reset | Fire Mode 1 | Fire Mode 2 | Fire Mode 3 | Fire Mode 4 |
|---|---|---|---|---|---|
| Accuracy (%) ↑ | 85.31 | **86.11** | 85.64 | 85.68 | 85.55 |
| Spiking Rate (%) ↓ | 12.27 | 13.07 | 12.19 | 12.19 | **11.94** |

Table 11: The impact of different fire modes for the fuzzy spiking signals on SPikE-SSM-Full with refractory period and trainable $U_{\mathrm{th}}$ and $V_{\mathrm{th}}$.

| Criterion | No Reset | Fire Mode 1 | Fire Mode 2 | Fire Mode 3 | Fire Mode 4 |
|---|---|---|---|---|---|
| Accuracy (%) ↑ | **85.45** | 85.34 | 84.80 | 84.84 | 85.23 |
| Spiking Rate (%) ↓ | 14.81 | 10.16 | 9.97 | 10.87 | **9.75** |

The results for SPikE-SSM-Full and SPikE-SSM-SR ($v_{th}$ and $U_{th}$ are trainable) are shown in Table 8 and Table 9 respectively. Note that there are two hyper-parameters ($\tau$ and $\tau_r$) in SPikE-SSM-Full, and only one hyper-parameter ($\tau$) in SPikE-SSM-SR. As previously stated by the default setting, the fuzzy spiking signals $s_f$ are all set to False during training (i.e. $s_f = 0$).

From Table 8, we can observe that: (1) In SPikE-SSM-SR with only one hyper-parameter $\tau$, regardless of the number of PMBC iterations, the change in $\tau$ has minimal impact on the model's overall accuracy and sparsity, indicating that the performance of SPikE-SSM-SR is not sensitive to the parameter $\tau$, which demonstrates the robustness of our proposed method. (2) As the number of PMBC iterations increases, the Fuzzy Rate significantly decreases, as more iterations in PMBC allow for more spiking signals to be identified. (3) Under the same number of PMBC iterations, the Fuzzy Rate decreases significantly as the hyper-parameter $\tau$ becomes smaller. This indicates that a smaller $\tau$ helps improve the computational efficiency of PMBC and the accuracy of the model.

From Table 9, we can observe that: (1) In SPikE-SSM-SR with two hyper-parameters $\tau$ and $\tau_r$, under the same number of PMBC iterations and $\tau_r$, the change in $\tau$ has minimal impact on the model's overall accuracy, but smaller $\tau$ leads to smaller Spiking Rate and Fuzzy Rate. This indicates that a smaller $\tau$ helps reduce the sparsity and improve the accuracy of the model. (2) Under the same number of PMBC iterations and $\tau$, the change in $\tau_r$ has minimal impact on the model's overall accuracy and sparsity, demonstrating the robustness of our method.

Therefore, the hyper-parameters $\tau$ and $\tau_r$ are set to 0.1 and 0.9 respectively in the full SPikE-SSMby default, which can achieve a more balanced performance between the sparsity and accuracy.

### C.3.2 THE IMPACT OF FUZZY FIRE MODES IN PMBC

Due to the parallel computation of PMBC, most of the spiking signals can be definitely obtained in 5-10 iterations of Algorithm 1 and Algorithm 2, with only a few spiking signals are still fuzzy. To improve computing efficiency, we adopt several fire modes to address these fuzzy spiking signals to reduce the iterations of PMBC, including:

- No Reset: Train the SPikE-SSM without reset mechanism (so $U_{\mathrm{th}}$ and refractory period are not applicable).
- Fire Mode 1: The fuzzy spiking signals $s_f$ are all set to True (i.e. $s_f = 1$).
- Fire Mode 2: The fuzzy spiking signals $s_f$ are all set to False (i.e. $s_f = 0$).
- Fire Mode 3: The fuzzy spiking signals $s_f$ randomly are determined by the mean spiking rate value of definite piking signals $s_d$.
- Fire Mode 4: The fuzzy spiking signals $s_f$ are determined by their current corresponding upper and lower bounds. For example, after all the iterations of PMBC, if $s_i$ is still fuzzy in time step $t = i$, and its upper and lower bounds are $m_i^{up}$ and $m_i^{low}$. If $k_i > (m_i^{up} + m_i^{low})/2$, then $s_i = 1$, or else $s_i = 0$.

Table 12: The impact of different numbers of iterations in PMBC on the accuracy and spiking rate performances of our method.

| Number of Iterations | 1 | 2 | 5 | 10 | 30 |
|---|---|---|---|---|---|
| Accuracy (%) ↑ | 84.46 | 84.74 | 84.74 | **85.02** | 84.99 |
| Spiking Rate (%) ↓ | 13.32 | 12.90 | 12.66 | **12.11** | 12.32 |
| Fuzzy Rate (%) ↓ | 8.80 | 6.77 | 4.48 | **2.62** | **0.82** |

Table 13: The impact of different numbers of iterations in PMBC on the speed and time cost of our method. "serial computing" means serially training SPikE-SSMwithout PMBC.

| Number of Iterations | 1 | 2 | 5 | 10 | 30 | 50 | serial computing |
|---|---|---|---|---|---|---|---|
| Speed (iters/s) ↑ | **25.32** | 21.51 | 14.63 | 9.50 | 3.97 | 2.42 | 1.21 |
| Time (ms/iters) ↓ | **39.49** | 46.49 | 68.35 | 105.26 | 251.89 | 413.22 | 826.45 |

To investigate the impact of different fire modes of fuzzy spiking signals for PMBC, we trained the SPikE-SSM-SR and SPikE-SSM-Full with different fire modes on dataset sCifar10 with 10 PMBC iterations and 100 epochs, and the performances are shown in Table 10 and Table 11 respectively.

According to Table 10, we can observe that SPikE-SSM-SR with Fire Mode 2, Fire Mode 3 and Fire Mode 4 can all achieve both higher accuracy and lower spiking rate than that of SPikE-SSMwithout the reset mechanism, which verifies the effectiveness of reset mechanism in the refractory LIF neuron model. Although SPikE-SSM-SR with Fire Mode 1 achieves the highest spiking rate, it also achieves the highest accuracy among the five versions, which can be attributed to the fact that too sparse signals may lead to the loss of important information during training.

According to Table 11, we can observe that SPikE-SSM-Full with Fire Mode 1, Fire Mode 2, Fire Mode 3, and Fire Mode 4 can all achieve lower spiking rates than that of SPikE-SSMwithout the reset mechanism. Notably, compared with Table 10 and Table 11, we can observe that SPikE-SSM-Full with both trainable $U_{th}$ and $v_{th}$ can achieve lower spiking rate than SPikE-SSM-SR with the same Fire Modes in all the four different Fire Modes, which indicates that the trainable $U_{th}$ and $V_{th}$ can more effectively model the temporal dynamics with biological interpretability.

In our method, we choose Fire Mode 2 as the default setting since it can further reduce the spiking rate with all the fuzzy spiking signals set to False ($s_f = 0$), and it can also achieve a more balanced performance between accuracy and spiking rate.

### C.3.3 THE IMPACT OF DIFFERENT ITERATIONS IN PMBC

In this section, we investigate the impact of different numbers of iterations in PMBC on the accuracy and sparsity performances of SPikE-SSM. We trained the SPikE-SSM-SR with fixed $U_{th} = v_{th} = 1$ and 4 layers on dataset sCIFAR10 with 100 epochs and $\tau = 0.2$. This experiment is conducted on a V-100 GPU. The results with different iterations in PMBC are shown in Table 12.

According to Table 12, we can observe that: (1) As the number of PMBC iterations increases, the model's accuracy gradually improves, while the fuzzy rate decreases. This is because more spiking signals are explicitly calculated with additional iterations, enhancing the model's robustness and leading to a steady rise in accuracy. (2) As the number of PMBC iterations increases, the overall spiking rate gradually decreases, indicating that more distinct spiking signals can enhance the model's sparsity. (3) The SPikE-SSM-SR with 10 iterations of PMBC achieves both higher accuracy and a lower spiking rate compared to the version with 30 iterations, despite having a significantly higher fuzzy rate. This suggests that more iterations of PMBC are not always better, as fuzzy spiking signals may function similarly to dropout.

In addition, we investigate the impact of different and more granular numbers of iterations in PMBC on the inference speed and the cost times of SPikE-SSM. The experimental settings are the same as Table 12. Note that the sequence length $L$ of sCIFAR10 is 1024. The experimental results are shown

in Table 13, from which we can observe that: (1) As the number of iterations in PMBC increases, the model's inference speed progressively decreases, with each iteration requiring more time to complete. This slowdown becomes more noticeable as iteration counts grow. (2) The fastest inference speed is achieved when only a single iteration is performed. However, even when multiple iterations are conducted in parallel within PMBC, the inference speed remains considerably faster than the traditional sequential iteration method that doesn't use PMBC. These findings further highlight the efficiency of our proposed PMBC-based training approach, demonstrating that it significantly accelerates the model's inference while maintaining robust performance.

In our method, we set the default number of iterations of PMBC as 3, which can achieve a more balanced performance between the high-efficiency inference and stable accuracy.

