# OpenReview forum: "SPikE-SSM: A Sparse, Precise, and Efficient Spiking State Space Model for Long Sequences Learning"
_ICLR.cc/2025/Conference — ICLR 2025 Conference Withdrawn Submission_

### Official Review · Reviewer_7BQs · 2024-10-16

**Soundness:** 2
**Presentation:** 2
**Contribution:** 1
**Rating:** 3
**Confidence:** 5

**Summary:**

The paper presents SPikE-SSM, a framework that aims to integrate spiking neural networks (SNNs) with state space models (SSMs) to address the challenges of long-sequence learning in an energy-efficient and sparse manner. The authors introduce several innovations such as the Parallel Max-Min Boundary Compression (PMBC) strategy for accelerating SNN inference, a refractory neuron model for temporal dynamics, and the integration of trainable thresholds for improved sparsity-accuracy trade-offs. Extensive experiments on LRA benchmarks and WikiText-103 demonstrate improved performance.

**Strengths:**

The proposed refractory neuron model is biologically inspired and brings interpretability to SNNs for long-sequence tasks. The experiments are well-structured, with results on a wide range of tasks (LRA benchmarks and WikiText-103) showing some promise.

**Weaknesses:**

The core contribution—integrating SNNs with SSMs—is somewhat incremental. There are previous works, such as SpikingSSM, that already explore similar ideas. The novelty of the proposed solutions does not clearly differentiate it from existing methods, especially since the claimed improvements (like PMBC) are marginally better in terms of accuracy. The paper proposes several assertions and claims without sufficient theoretical backing. The proofs in the appendices are not rigorous enough and seem more heuristic than formal. This weakens the impact of the PMBC algorithm, which is a major part of the contribution. While the authors test their approach on multiple benchmarks, the results do not show significant improvements over previous methods. In many cases, the accuracy gains are minimal (sometimes less than 1%) compared to competing models, particularly when considering the energy-efficiency trade-offs. Furthermore, the benchmarks chosen (such as sMNIST) are relatively simple, and the performance on more challenging real-world tasks could have been more extensively evaluated.

One of the key selling points of SNNs is their energy efficiency. However, while the authors present some energy consumption estimates, these are based on theoretical assumptions (e.g., number of operations) rather than real-world implementations on neuromorphic hardware. Without real hardware measurements, the claimed energy benefits are speculative and diminish the practical relevance of the paper. The paper is dense and difficult to follow, particularly in sections where the PMBC algorithm and the refractory neuron model are introduced. The text often lacks clarity, and it is not always clear how the various components fit together. This makes the methodology hard to replicate.

**Questions:**

Detailed questions you can refer to weaknesses.

**Details Of Ethics Concerns:**

The paper presents an incremental improvement over existing SNN-SSM methods but lacks the novelty, theoretical rigor, and experimental results to justify acceptance. While the proposed PMBC strategy and refractory neuron model are interesting, their practical benefits are not convincingly demonstrated. Additionally, the paper’s clarity issues further detract from its overall impact. A stronger focus on theoretical backing, real-world applications, and improved clarity would be needed for future consideration.

---

### Official Review · Reviewer_TMr3 · 2024-10-27

**Soundness:** 2
**Presentation:** 2
**Contribution:** 2
**Rating:** 3
**Confidence:** 5

**Summary:**

This paper proposes a spiking neural network model based on SSM. It employs a method similar to fixed-point iteration to address the challenge of training the reset mechanism in parallel. Additionally, the model introduces a refractory period for neurons, enhancing biological interpretability and resulting in sparser spike patterns.

**Strengths:**

1. A method similar to fixed-point iteration is proposed to solve the output based on the input sequence efficiently. This method can run in parallel, accelerating the training process.
2. Better accuracy and sparsity compared with other spikingSSMS.

**Weaknesses:**

1. In Figure 1, the input to the GLU is a floating-point number. Is the multiplication of floating-point numbers unavoidable in this case? Or is there a more effective way to replace this module?
2. While the article introduces sparsity through the reset mechanism and refractory period, is this nonlinear transformation interpretable? Specifically, how do the reset modules and refractory periods enhance model performance?

**Questions:**

1. In Figure 1, the input to the GLU is a floating-point number. Is the multiplication of floating-point numbers unavoidable in this case? Or is there a more effective way to replace this module?
2. While the article introduces sparsity through the reset mechanism and refractory period, is this nonlinear transformation interpretable? Specifically, how do the reset modules and refractory periods enhance model performance?

---

### Official Review · Reviewer_pRy7 · 2024-11-02

**Soundness:** 3
**Presentation:** 2
**Contribution:** 3
**Rating:** 5
**Confidence:** 5

**Summary:**

The paper presents SPikE-SSM, a novel spiking state space model designed to address key challenges in long-sequence learning with spiking neural networks (SNNs). The authors introduce a boundary compression strategy (PMBC) to accelerate spiking neuron model inference, enabling parallel processing for long sequence learning. Additionally, they propose a new LIF neuron model with a reset-refractory mechanism to exploit the temporal dimension for biologically interpretable dynamic computation. The model's evaluation on long-range arena benchmarks and the large language dataset WikiText-103 demonstrates the potential of dynamic spiking neurons for efficient long-sequence learning.

**Strengths:**

1. The introduction of the PMBC strategy enables parallel processing in SNNs, representing an innovative advancement. Additionally, the proposed LIF neuron model incorporates a reset-refractory mechanism, enhancing both biological interpretability and dynamic computational capabilities.
2. The article is well-structured, making the authors' arguments and experimental content easy to follow. The experimental results partially demonstrate the suitability of the proposed method as an SNN architecture for long-sequence tasks.

**Weaknesses:**

1. The scalability of SPikE-SSM to larger datasets and more complex tasks has not been thoroughly discussed. Additionally, there is a lack of comparison with non-SSM-based SOTA SNN architectures in terms of computational efficiency and energy consumption.
2. The presentation of PMBC in Figure 2 is unclear, making it difficult for readers to grasp the design rationale and core concepts being conveyed.
3.What does "x,h,x,...,x" represent in Figure 3? It is not explained in the caption, which may lead to confusion.
4. I couldn't find a complete diagram of the network architecture, nor any information on parameters or settings, in either the main text or the appendix.
5. To my knowledge, there are several existing strategies for parallel training of SNNs[1][2], which the authors did not compare in this paper. What are the advantages of the proposed approach compared to these existing methods?
[1] Fang, Wei, et al. "Parallel spiking neurons with high efficiency and ability to learn long-term dependencies." Advances in Neural Information Processing Systems 36 (2024).
[2] Zhang, Shimin, et al. "Tc-lif: A two-compartment spiking neuron model for long-term sequential modelling." Proceedings of the AAAI Conference on Artificial Intelligence. Vol. 38. No. 15. 2024.

**Questions:**

1. The author's point-by-point discussion is well-executed but lacks an explanation of the logical relationships between the three different issues, which makes them feel somewhat disconnected. What is the logical relationship among these three issues—is it progressive or parallel?

---

### Note · Authors · 2024-11-26

I have read and agree with the venue's withdrawal policy on behalf of myself and my co-authors.